# Sex and Age Differences in a Progressive Synucleinopathy Mouse Model

**DOI:** 10.3390/biom13060977

**Published:** 2023-06-11

**Authors:** Jérôme Lamontagne-Proulx, Katherine Coulombe, Marc Morissette, Marie Rieux, Frédéric Calon, Thérèse Di Paolo, Denis Soulet

**Affiliations:** 1Centre de Recherche du CHU de Québec, Axe Neurosciences, T2-32, 2705, Boulevard Laurier, Québec, QC G1V 4G2, Canada; jerome.lamontagne-proulx@vrr.ulaval.ca (J.L.-P.);; 2Laboratoire International Associé OptiNutriBrain (NutriNeuro France-INAF Canada), Québec, QC G1V 0A6, Canada; 3Institut sur la Nutrition et les Aliments Fonctionnels (INAF), Université Laval, 2440, Boulevard Hochelaga, Bureau 1705, Québec, QC G1V 0A6, Canada

**Keywords:** Parkinson’s disease, sex differences, synucleinopathy models, dopaminergic neuron, SNCA-OVX, alpha-synuclein

## Abstract

The mutation and overexpression of the alpha-synuclein protein (αSyn), described as synucleinopathy, is associated with Parkinson’s disease (PD)-like pathologies. A higher prevalence of PD is documented for men versus women, suggesting female hormones’ implication in slowing PD progression. The nigrostriatal dopamine (DA) neurons in rodent males are more vulnerable to toxins than those in females. The effect of biological sex on synucleinopathy remains poorly described and was investigated using mice knocked out for murine αSyn (SNCA-/-) and also overexpressing human αSyn (SNCA-OVX) compared to wildtype (WT) mice. All the mice showed decreased locomotor activity with age, and more abruptly in the male than in the female SNCA-OVX mice; anxiety-like behavior increased with age. The SNCA-OVX mice had an age-dependent accumulation of αSyn. Older age was associated with the loss of nigral DA neurons and decreased striatal DA contents. The astrogliosis, microgliosis, and cytokine concentrations increased with aging. More abrupt nigrostriatal DA decreases and increased microgliosis were observed in the male SNCA-OVX mice. Human αSyn overexpression and murine αSyn knockout resulted in behavioral dysfunctions, while only human αSyn overexpression was toxic to DA neurons. At 18 months, neuroprotection was lost in the female SNCA-OVX mice, with a likely loss of estrus cycles. In conclusion, sex-dependent αSyn toxicity was observed, affecting the male mice more significantly.

## 1. Introduction

The SNCA gene, encoding for alpha-synuclein protein (αSyn), was the first gene associated with familial forms of Parkinson’s disease (PD) [1,2]. Since this discovery, the overexpression and accumulation of αSyn in the brain has been considered as a hallmark of both SNCA-based familial and sporadic forms of PD [3,4]. The accumulation of this protein into insoluble fibrillar forms leads to the formation of neuronal inclusion in cell bodies and neurites, referred to as Lewy bodies and Lewy neurites, respectively [5,6]. The αSyn is a 140-amino-acid protein mainly found in cytosol or bound to membranes at certain presynaptic terminals [7,8]. The functions of αSyn are still debated, although evidence points to its potential to modulate the fusion pore of the membrane via its role in soluble N-ethylmaleimide-sensitive-factor attachment protein receptor (SNARE)-complex formation. Previous studies indicated that the overexpression and the loss of endogenous αSyn inhibits exocytosis [9,10,11]. Other research reports showed that the protein can travel anterogradely and retrogradely in neurons using axonal transport and can be released in extracellular environments [12,13]. Moreover, the prion-like spread of αSyn pathology between infected and healthy cells has been well documented in the last decade [14,15,16,17].

Inclusions of αSyn are mainly found in the substantia nigra pars compacta (SNpc), where the loss of dopamine (DA) neurons and axonal fibers is a typical feature in PD [18]. The implication of αSyn accumulation in neurodegeneration is not fully understood, but evidence suggests that this protein could contribute to the dysfunction of neurons in both the central nervous system (CNS) and the enteric nervous system (ENS), classifying PD as a synucleinopathy [19,20]. Furthermore, αSyn has been reported to decrease DA release and increase pro-inflammatory responses through sustained microgliosis, astrogliosis, and, subsequently, elevated cytokine production [21,22,23]. Recently, a study showed that an increase in αSyn aggregates and a loss of DA-neuron integrity was associated with motor deficits in a mutated αSyn model of PD. Interestingly, the neuropathology was significantly delayed in the female compared to the male mice, and an estrogen treatment mitigated the PD-like pathology in the model [24].

Sex differences are major factors influencing the incidence, course, and nature of a large number of disorders, including neurodevelopmental and neurodegenerative diseases [25]. Moreover, PD is not an exception, since its prevalence in men is 1.5 higher than women [26,27,28]. Furthermore, the reported age of diagnosis is different for the two sexes, with an average of 51.3 years in men and 53.4 years in women [29]. In terms of the symptoms, early and persistent tremor, constipation, fatigue, pain, depression, and dyskinesias are more prevalent in women, as opposed to the higher occurrence of rigidity, daytime sleepiness, drooling, and sexual dysfunction in male patients [29,30]. Estrogens appear to be major factors in explaining the sex differences in PD. Epidemiological studies reported that women affected with PD experienced an earlier end to their reproductive lifespan and that the increase in menopausal age reduces the risk of PD [31,32]. Furthermore, many studies reported the efficacy of estrogen-receptor modulators or estrogen-receptor agonists in decreasing the extent of the pathology in animal models of PD [33,34,35]. The effects of these sex differences are well documented in idiopathic PD patients and neurotoxin-based animal models of PD. However, there is little evidence supporting the sex-dependent evolution of the pathology in familial forms of PD and progressive genetic animal models [36]. Consequently, elucidating the implication of sex in αSyn-related neuropathology can not only lead to a better understanding of the role of this protein in PD, but may also provide insights into the significance of sex differences in the SNCA-based genetic forms of the disease. In this study, we used SNCA-OVX mice, which overexpress human αSyn (h-αSyn) under its endogenous promoter. The characteristics of this model were previously described showing a 1.9-fold higher expression of the SNCA gene than the endogenous αSyn in C57/Bl6 mice [37]. In addition, these mice exhibited various phenotypes, such as motor deficits, loss of nigral DA neurons, and striatal DA impairment [37]. Therefore, this model can provide a valuable tool with which to study biological sex differences in the development of αSyn-associated diseases. The accumulation of αSyn causes progressive motor impairment and cognitive deficits, and the hypothesis is that female and male SNCA-OVX mice display age-related sex differences in the manifestation of these phenotypes (behavior, DA systems, and neuroinflammation). Hence, the aim of the present study was to compare overexpressing αSyn mice behaviorally and biochemically in terms of the effect of sex and disease progression during aging. This animal model can help to understand the role of sex and sex hormones in the sequence of events leading to PD, which could help to decipher the differences observed between women and men suffering from PD.

## 2. Materials and Methods

### 2.1. Animals

The SNCA-OVX line (Tg (SNCA)OVX37Rwm) was generated by injection of a bacterial artificial chromosome (BAC) DNA containing the human SNCA locus into C57BL/6 mice and then backcrossed with SNCA-/- mice for 9 generations, as provided by Richard Wade-Martins (JAX Genomic Medicine, Farmington, CT, USA) [37]. At Jackson Laboratory, SNCA-OVX males were backcrossed with SNCA-/- females and sent to our animal facility. Female and male SNCA-OVX, SNCA-/-, and C57BL/6 (wildtype) mice were bred in our animal facility. Animals were housed in ventilated cages, on a 12-h/12-h dark/light cycle. All experiments were approved by the animal research committee of the Centre de recherche du CHU de Québec—Université Laval (#18-039) and performed according to the Canadian Guide for the Care and Use of Laboratory Animals. In total, 90 mice were obtained for 4-, 12-, and 18-month-old groups, respectively, thus giving 10 mice per sex and genotype.

### 2.2. Weight, Open Field Analysis, and Stool Collection

Weight measurements, open field tests, and stool collection were performed longitudinally with the 18-month-old group (called group 18). Weights were measured every month from 2 to 18 months of age. Open field testing was performed at the age of 4, 12, and 18 months for all mice in group 18 (Figure 1A). They were allowed to move freely in a box with an area of 1 m^2^ for 10 min and were tracked using a customized program written in MATLAB 2018b environment (MathWorks, Natick, MA, USA). Locomotor activity as the distance travelled and anxiety-like behavior as percentage of time in the inner zone were measured. Stools harvested for 1 h were weighed, heated at 60 °C overnight under ventilation, and weighed again, in order to measure the percentage of water and the number of stools.

### 2.3. Tissue Preparation

Animals were euthanized at 4, 12, or 18 months of age via intracardiac perfusion of 1X PBS (Figure 1A). The blood was collected and centrifuged at 2500× *g* for 15 min, and then plasma was harvested and stored at −80 °C. The brains were collected and separated in two hemispheres. The right hemisphere was post-fixed in 4% paraformaldehyde (PFA) for 48 h, and then stored in a solution composed of 20% sucrose and 0.05% sodium azide in 1X PBS pH 7.4 at 4 °C. From this hemisphere, 30-μm-thick sections were prepared using a sliding microtome (Leica Microsystems Canada Inc., Richmond Hill, ON, Canada) and stored in cryoprotectant solution for further immunofluorescence analysis. The left hemisphere was snap-frozen in isopentane on dry ice (−40 °C) and the anterior part of the striatum was dissected for biogenic amine assay.

### 2.4. Immunofluorescence

Free-floating brain sections were incubated for 1 h at 100 °C in sodium citrate for antigen retrieval before a 30-min treatment with a blocking solution of 0.4% Triton X-100 and 5% donkey serum (Sigma-Aldrich Canada, Oakville, ON, Canada) in 1X PBS. Tissues were stained overnight with primary antibodies, followed by a 2-h incubation with secondary antibodies, both in blocking solution. Counterstaining with 0.022% DAPI (Invitrogen Corporation, Waltham, MA, USA) was performed before mounting sections. Refer to Appendix A for the list of antibodies. Brain sections were imaged using a Zeiss AxioScan.Z1 scanner and Zen 3.1 acquisition software (Carl Zeiss Canada, Toronto, ON, Canada). Mean pixel intensity was determined on images for immunofluorescence quantification, and for density count, labeled cells were calculated as the number of positive cells per area (mm^2^), using Fiji software [38]. Mean values were calculated with 7–10 sections for each animal. All images and data analyses were conducted blindly.

### 2.5. Striatal Biogenic Amine Assay

The left anterior striata were dissected (approximately 1.54 mm to 0.62 mm from bregma), homogenized in 250 μL of 0.1 N HClO_4_ at 4 °C and then centrifuged at 10,000× *g* for 30 min (4 °C) to precipitate proteins. The striatal contents of DA and its metabolites, 3,4-dihydroxyphenylacetic acid (DOPAC), 3-methoxytyramine (3-MT), and homovanillic acid (HVA), as well as serotonin (5-HT) and its metabolite 5-hydroxyindoleacetic acid (5-HIAA), were measured by high-performance liquid chromatography (HPLC) with electrochemical detection. Supernatants of striatal tissue were directly injected into the chromatograph consisting of a Waters 717 plus autosampler automatic injector, a Waters 515 pump equipped with a C-18 column (Waters Nova-Pak C18, 3 μm, 3.9 mm × 150 cm, Waters Corp., Milford, MA, USA), a BAS LC-4C electrochemical detector, and a glassy carbon electrode. The mobile phase consisted of 0.025 M citric acid, 1.7 mM 1-heptane-sulfonic acid, and 10% methanol, in filtered distilled water, delivered at a flow rate of 0.8 mL/min. The final pH of 3.98 was obtained by addition of NaOH. The electrochemical potential was set at 0.8 V with respect to an Ag/AgCl reference electrode. Proteins were assayed with a Micro BCA protein assay kit (Thermo Scientific, Waltham, MA, USA) and results were expressed in nanograms of amine per milligram of protein.

### 2.6. Cytokine Immunoassay

Cytokine levels in plasma of animals were quantified using a Bio-Plex Pro Mouse Cytokine 8-Plex assay kit (#M60000007A, BioRad, Hercules, CA, USA) according to manufacturer’s instructions. In total, 25 μL of each diluted (1:2) plasma from samples and 12.5 μL of diluted (1:4) plasma from 2 lipopolysaccharide (LPS)-treated mice (positive control) were used for immunoassay analysis. Standard curves were used to determine IL-1β, IL-4, IL-5, IL-10, IFN-γ, and TNF-α values and final calculation was performed according to the dilution. The IL-2 and GM-CSF were not detectable in any of the samples.

### 2.7. Statistical Analysis

Data were evaluated with the Shapiro–Wilk test to determine normality and outliers were discarded with the interquartile range method. Comparisons of weights between females and males were analyzed with an unpaired t-test. Comparisons between groups with sexes separated were analyzed with a two-way ANOVA analysis followed by Bonferroni post hoc tests for each mouse model (with age and sex as independent variables). Comparisons between groups with pooled sexes were calculated by two-way ANOVA analysis followed by Bonferroni post hoc tests (with genotype and age as independent variables). Comparisons between time points of animals of same sex and genotype were calculated by one-way ANOVA, followed by Tukey’s post hoc tests. Moreover, comparisons between sexes of animals of the same genotype and the same time point were calculated by a one-way ANOVA followed by Sidak post hoc tests. Results were represented as the mean ± SEM of 6–10 mice per group. Results were considered statistically significant if *p* < 0.05. All statistical analyses were performed with Prism 9.5.1 (GraphPad Software Inc., La Jolla, CA, USA) software.

## 3. Results

### 3.1. Effect of Age and Sex on Mouse Weights

The female and male SNCA-OVX, SNCA-/-, and wildtype mice were bred to generate two groups. One was euthanized at 4 months of age to model an early stage of the disease and the second was submitted to a series of longitudinal behavioral analyses and feces collections at 4 and 12 months and before the euthanasia at 12 and 18 months (Figure 1A). The 18-months group was weighed each month, revealing progressive weight gain between 2 and 18 months. As expected, the males were significantly heavier than the females at each timepoint (Figure 1B).

### 3.2. Effects of Age and Sex on Motor and Non-Motor Deficits

The animals were exposed to the open field test at 4, 12, and 18 months of age to analyze the progressive locomotor dysfunctions and anxiety-like behavior, as previously described [39,40]. The locomotor activity was measured by the total distance travelled in 10 min, and the results revealed a decrease in motor activity in the wildtype mice with aging (age factor F (2.26) = 11.48, *p* = 0.0004) and a lower activity level in the males than in the females (sex factor F (1.17) = 5.06, *p* = 0.0380). The SNCA-/- mice also showed a progressive decrease in motor activity with aging (age factor F (2.32) = 30.38, *p* < 0.0001), but without differences between their biological sexes (sex factor F (1.18) = 0.28, *p* = 0.6017). The SNCA-OVX mice showed a progressive decrease in the total distance travelled (age factor F (2.23) = 19.35, *p* < 0.0001) and a difference between females and males (sex factor F (1.18) = 8.86, *p* = 0.0081) (Figure 1C).

The anxiety-like behavior was measured by the decrease in the time mice spent in the inner zone of the open field test. The wildtype mice showed a small progressive development of anxiety (age factor F (2.31) = 8.39, *p* = 0.0014), but without sex differences (sex factor F (1.17) = 0.37, *p* = 0.5507). Furthermore, the SNCA-/- and SNCA-OVX mice both showed a significant progressive increase in anxiety-like behavior (SNCA-/-: age factor F (2.29) = 32.63, *p* < 0.0001; SNCA-OVX: age factor F (2.33) = 11.97, *p* = 0.0001), with no differences between the males and females (SNCA-/-: sex factor F (1.18) = 1.52, *p* = 0.2330; SNCA-OVX: sex factor F (1.18) = 1.38, *p* = 0.2556) (Figure 1D). Overall, these behavioral results suggest that the main effect of *α*Syn overexpression is further deterioration in terms of motor and anxiety-like behavior in mice, beyond that which is normally observed with age. The motor behavior, but not anxiety-like activity, showed a more abrupt decline in the male versus the female SNCA-OVX mice.

Gastrointestinal dysfunction was observed in the early stages of the disease and affected more than 50% of the PD patients [41]. We performed a set of analyses by collecting fresh stools for one hour, which were counted and dried overnight to calculate the proportion of water, thus allowing us to obtain two indirect measures of intestinal motility. The results did not show changes in the proportion of water in either of the transgenic models (SNCA-/-: age factor F (2.27) = 2.31, *p* = 0.1232; SNCA-OVX: age factor F (2.30) = 2.42, *p* = 0.1108), but small variations in the wildtype mice that were not progressive (age factor F (2.41) = 8.34, *p* = 0.0012) were observed (Appendix A). Furthermore, the number of stools excreted per hour increased with age only in the wildtype male and SNCA-OVX female mice (wildtype: age factor F (2.31) = 10.90, *p* = 0.0003; SNCA-OVX: age factor F (2.33) = 3.60, *p* = 0.0385), and not in any of the SNCA-/- mice (age factor F (2.33) = 0.11, *p* = 0.8891) (Appendix A). Thus, based on the proportions of water analyzed (wildtype: sex factor F (1.45) = 29.32, *p* < 0.0001; SNCA-/-: sex factor F (1.18) = 13.33, *p* = 0.0018; SNCA-OVX: sex factor F (1.18) = 10.30, *p* = 0.0049) and the stool counts (wildtype: sex factor F (1.17) = 14.92, *p* = 0.0013; SNCA-/-: sex factor F (1.18) = 3.04, *p* = 0.0985; SNCA-OVX: sex factor F (1.18) = 3.18, *p* = 0.0916), the female mice seemed to have different levels of intestinal transit from those of the male mice, regardless of the genotype (Appendix A).

### 3.3. Accumulation of Human αSyn and Loss of Nigral DA Neurons in the SNCA-OVX Mice

To confirm the expression and the accumulation of h-αSyn in the SNCA-OVX model, we analyzed the immunoreactivity of a specific h-αSyn antibody in the striata of all the mice. Prior to the analyses, we established a baseline for the background immunoreactivity of this antibody, using five randomly chosen 18-month-old female and male SNCA-/- mice. The analysis of the h-αSyn showed an increase in the protein with aging in the SNCA-OVX mice (SNCA-OVX: age factor F (1.30) = 5.10, *p* = 0.0313; Figure 2A(i,ii)). As expected, the levels of h-αSyn remained near the baseline for the wildtype and SNCA-/- mice of all ages, resulting in a significant difference from the SNCA-OVX mice (genotype factor F (2.94) = 806.1, *p* < 0.0001, Figure 2A(ii)). Interestingly, the female SNCA-OVX mice had a trend of higher levels of h-αSyn than the males in both age groups (SNCA-OVX: sex factor F (1.30) = 3.34, *p* = 0.0775; Figure 2A(i)).

The loss of nigrostriatal DA neurons is a hallmark of neuropathological changes in the brains of PD patients and can be observed in most models [42,43,44,45,46,47]. Therefore, we investigated the integrity of the nigrostriatal pathway using tyrosine hydroxylase (TH), a rate-limiting enzyme in the synthesis of DA, as a marker for DA neurons. The quantification of the density of the TH+ neurons bodies in the SNpc revealed a loss of nigral DA neurons in the SNCA-OVX mice between 4 and 18 months of age (wildtype: age factor F (1.29) = 0.07, *p* = 0.7906; SNCA-/-: age factor F (1.30) = 2.81, *p* = 0.1039; SNCA-OVX: age factor F (1.28) = 10.54, *p* = 0.0030; Figure 2B(i)). This considerable age-dependent loss was upheld in the sex-pooled SNCA-OVX population, while there were no significant changes between genotypes (genotype factor F (2.93) = 1.19, *p* = 0.3101, Figure 2B(ii)). The TH labelling for the DA projection in the striatum showed non-significant changes with aging (wildtype: age factor F (1.30) = 0.98, *p* = 0.3293; SNCA-/-: age factor F (1.31) = 0.02, *p* = 0.8779; SNCA-OVX: age factor F (1.31) = 1.97, *p* = 0.1709; Figure 2C(i)) and no trends between genotypes (genotype factor F (2.98) = 2.98, *p* = 0.0552, Figure 2C(ii)). Moreover, no sex differences were observed in the DA neurons of either the SNpc (wildtype: sex factor F (1.29) = 0.09, *p* = 0.7610; SNCA-/-: sex factor F (1.30) = 0.70, *p* = 0.4101; SNCA-OVX: sex factor F (1.28) = 0.91, *p* = 0.3481; Figure 2B(i)) or the striatum (wildtype: sex factor F (1.30) = 0.003, *p* = 0.9551; SNCA-/-: sex factor F (1.31) = 0.20, *p* = 0.6544; SNCA-OVX: sex factor F (1.31) = 0.0006, *p* = 0.9936; Figure 2C(i)). Taken together, our results suggest that the neurodegeneration of nigral DA neurons in the SNCA-OVX mice could have been a consequence of h-*α*Syn accumulation.

### 3.4. Aging of SNCA-OVX Mice g Resulted in a Progressive Sex-Dependent Loss of Striatal Biogenic Amines

To examine the effects of the loss of DA neurons in the nigrostriatal pathway, we quantified the levels of striatal biogenic DA and 5-HT, as well as their metabolites. The analysis of the DA revealed a decrease in the neurotransmitter with aging only in the SNCA-OVX mice (wildtype: age factor F (2.34) = 0.22, *p* = 0.8037; SNCA-/-: age factor F (2.38) = 0.84, *p* = 0.4416; SNCA-OVX: age factor F (2.36) = 15.80, *p* < 0.0001; Figure 3A(i)). Interestingly, the striatal DA content in the 4-month-old SNCA-OVX mice was higher than that in their 18-month counterparts, and it was also higher than in the other genotypes (genotype factor F (2.117) = 13.60, *p* < 0.0001; Figure 3A(ii)). Notably, the reduction in the striatal DA content was already significant at 12 months in the male SNCA-OVX mice, while the loss was more progressive and non-significant at 12-months in the female SNCA-OVX mice, suggesting a sex difference in the disease’s progression. We also observed a decrease in DA metabolites with aging in the SNCA-OVX mice: DOPAC (wildtype: age factor F (2.34) = 1.19, *p* = 0.3173; SNCA-/-: age factor F (2.38) = 0.27, *p* = 0.7641; SNCA-OVX: age factor F (2.37) = 9.52, *p* = 0.0005; Figure 3B(i)), 3-MT (wildtype: age factor F (2.33) = 1.74, *p* = 0.1920; SNCA-/-: age factor F (2.35) = 13.47, *p* < 0.0001; SNCA-OVX: age factor F (2.36) = 13.93, *p* < 0.0001; Figure 3C(i)) and HVA (wildtype: age factor F (2.32) = 1.32, *p* = 0.2825; SNCA-/-: age factor F (2.37) = 0.08, *p* = 0.9220; SNCA-OVX: age factor F (2.34) = 8.06, *p* = 0.0014, Figure 3D(i)). Along with the high level of DA in the 4-month-old SNCA-OVX mice, the levels of DA metabolites were higher in this group compared with the other mice (DOPAC genotype factor F (2.117) = 4.96, *p* = 0.0086, Figure 3B(ii); 3-MT genotype factor F (2.113) = 8.22, *p* = 0.0005, Figure 3C(ii); HVA genotype factor F (2.112) = 3.78, *p* = 0.0257, Figure 3D(ii)). The comparison of the striatal catecholamine contents of all the experimental groups showed significant differences in DA (F (17.109) = 5.31, *p* < 0.0001), DOPAC (F (17.111) = 3.642, *p* < 0.0001), 3-MT (F (17.111) = 5.216, *p* < 0.0001), and HVA (F (17.111) = 3.938, *p* < 0.0001) contents, and the post hoc comparisons of sex differences showed higher DA contents in the female compared to the male SNC-OVX mice at 12 months of age (Figure 3A(i)), but not in DOPAC (Figure 3B(i)) or HVA (Figure 3D(i)). The striatal 3-MT contents were higher in the female wildtype mice at 4 and 18 months and in the SNCA-OVX females at 4 months of age compared to the males of these genotypes at the same age Figure 3C(i).

Since serotonergic alterations within the striata of PD patients can be related to non-motor symptoms, such as anxiety, we measured the levels of 5-HT and its metabolites 5-HIAA [48]. Furthermore, a previous study showed a significant decrease in the release of 5-HT in a mouse model with transgenic αSyn [49]. Our results indicate that there was no decrease in striatal 5-HT content with aging (wildtype: age factor F (2.32) = 0.23, *p* = 0.7994; SNCA-/-: age factor F (2.38) = 1.92, *p* = 0.1606; SNCA-OVX: age factor F (2.36) = 3.03, *p* = 0.0609, Figure 3E(i)), and that there were no differences between the genotypes (genotype factor F (2.115) = 1.13, *p* = 0.3265, Figure 3E(ii)). No differences in 5-HIAA content were observed with aging (wildtype: age factor F (2.33) = 0.15, *p* = 0.8652; SNCA-/-: age factor F (2.35) = 0.24, *p* = 0.7860; SNCA-OVX: age factor F (2.34) = 0.53, *p* = 0.5928; Figure 3F(i)) or between genotypes (genotype factor F (2.111) = 1.80, *p* = 0.1696, Figure 3F(ii)). A comparison of the striatal indoleamine contents of all the experimental groups showed significant differences in 5-HT (F (17.109) = 2.489, *p* = 0.00240) and 5-HIAA (F (17.109) = 2.220, *p* = 0.0071) contents, and post hoc comparisons of sex differences showed higher 5-HT contents in the male compared to the female wildtype mice at 12 months of age (Figure 3E(i)). The striatal 5-HIAA contents in the female SNCA-OVX genotypes were higher than those in the males at 12 months of age (Figure 3F(i)). These findings suggest the loss of nigrostriatal DA neurons and locomotor dysfunction in SNCA-OVX mice.

### 3.5. Effects of Aging, Sex, and Accumulation of Human αSyn in Brain Inflammation

Multiple lines of evidence suggest that the activation of the immune system could play an essential role in neurodegeneration [50]. While pro-inflammatory processes are increased naturally with aging, the aggregation of *α*Syn may induce a stronger immune response [51,52,53]. Our analysis of astrogliosis through the immunoreactivity of the glial fibrillary acidic protein (GFAP) in the striatum showed an increase with aging in the females of all the genotypes. However, we also observed an increase in astrogliosis in the 18-month-old male mice compared to the 4-month-old mice, but only in the SNCA-OVX group (wildtype: age factor F (1.31) = 10.42, *p* = 0.0029; SNCA-/-: age factor F (1.31) = 11.93, *p* = 0.0016; SNCA-OVX: age factor F (1.31) = 15.83, *p* = 0.0004; Figure 4A(i)). Statistically, no differences were observed between the genotypes (genotype factor F (2.99) = 2.87, *p* = 0.0613, Figure 4A(ii)) or the sexes (wildtype: sex factor F (1.31) = 1.71, *p* = 0.2012; SNCA-/-: sex factor F (1.31) = 0.28, *p* = 0.6032; SNCA-OVX: sex factor F (1.31) = 0.18, *p* = 0.6762; Figure 4A(i)). Moreover, we also quantified the microgliosis through the immunoreactivity of the ionized calcium-binding adapter molecule (Iba1) in the striatum and found a general increase in the marker in all the 18-month-old mice compared to the 4-month-old animals. However, the differences were only significant in the male SNCA-OVX mice (wildtype: age factor F (1.28) = 3.05, *p* = 0.0917; SNCA-/-: age factor F (1.30) = 4.82, *p* = 0.0360; SNCA-OVX: age factor F (1.31) = 4.37, *p* = 0.0448; Figure 4B(i)). No differences were observed between the genotypes (genotype factor F (2.95) = 2.91, *p* = 0.0595, Figure 4B(ii)) or the sexes (wildtype: sex factor F (1.28) = 0.25, *p* = 0.6228; SNCA-/-: sex factor F (1.30) = 0.17, *p* = 0.6820; SNCA-OVX: sex factor F (1.31) = 0.13, *p* = 0.7175; Figure 4B(i)). Both analyses indicated an increase in astrogliosis and microgliosis with aging, and only the SNCA-OVX male mice showed a higher increase in pro-inflammatory responses than the female mice.

Next, we investigated the levels of different cytokines (IL-1β, IL-5, IL-10, IFN-γ) in the plasma to confirm the increase in inflammation with aging and in the SNCA-OVX mice. The IL-1β levels were below the limit of detection for the 4-month-old mice. Even without the possibility of performing a statistical analysis for the age factor, it appeared that there was an increase in IL-1β with aging (Figure 4C(i)). Furthermore, the differences between the genotypes suggest a higher increase in this cytokine with aging in the SNCA-OVX and SNCA-/- mice compared to the wildtype mice (genotype factor F (2.37) = 10.36, *p* = 0.0003, Figure 4C(ii)). No differences between sexes were observed (sex factor F (1.37) = 0.34, *p* = 0.5613, Figure 4C(i)). The IL-5 levels were not significantly increased with aging in the female SNCA-/- or the female SNCA-OVX mice (wildtype: age factor F (1.22) = 4.54, *p* = 0.0445; SNCA-/-: age factor F (1.19) = 3.53, *p* = 0.0757; SNCA-OVX: age factor F (1.23) = 3.00, *p* = 0.0969; Figure 4D(i)), leading to a difference between the genotypes (genotype factor F (2.70) = 5.85, *p* = 0.0045, Figure 4D(ii)) and sex differences in both transgenic models (wildtype: sex factor F (1.22) = 2.56, *p* = 0.1240; SNCA-/-: sex factor F (1.19) = 6.39, *p* = 0.0205; SNCA-OVX: sex factor F (1.23) = 12.14, *p* = 0.0020; Figure 4D(i)). The IL5 levels were higher in the female compared to the male SNC-OVX mice at 4 and 18 months and at 18 months in the SNCA-/- mice (F (11.64) = 5.58, *p* < 0.001). The IL-10 quantification also showed an increase with aging in all the genotypes (wildtype: age factor F (1.24) = 52.95, *p* < 0.0001; SNCA-/-: age factor F (1.20) = 28.47, *p* < 0.0001; SNCA-OVX: age factor F (1.23) = 35.39, *p* < 0.0001; Figure 4E(i)). However, the levels of IL-10 were lower in both transgenic models compared to the wildtype (genotype factor F (2.73) = 7.48, *p* = 0.0011, Figure 4E(ii)) and lower in the male SNCA-OVX compared to the female SNCA-OVX (wildtype: sex factor F (1.24) = 0.03, *p* = 0.8695; SNCA-/-: sex factor F (1.20) = 0.19, *p* = 0.6714; SNCA-OVX: sex factor F (1.23) = 4.57, *p* = 0.0433; Figure 4E(i)). The IFN-γ levels were increased with aging in all the groups (wildtype: age factor F (1.23) = 17.91, *p* = 0.0003; SNCA-/-: age factor F (1.23) = 17.05, *p* = 0.0004; SNCA-OVX: age factor F (1.25) = 15.50, *p* = 0.0006; Figure 4F(i)), whereas the differences between the genotypes (genotype factor F (2.77) = 2.08, *p* = 0.1318, Figure 4F(ii)) and the sexes (wildtype: sex factor F (1.23) = 0.01, *p* = 0.9258; SNCA-/-: sex factor F (1.23) = 0.40, *p* = 0.5321; SNCA-OVX: sex factor F (1.25) = 0.72, *p* = 0.4028; Figure 4F(i)) were not significant.

The quantification of the IL-4 showed no statistical differences with aging between the groups (wildtype: age factor F (1.22) = 0.07, *p* = 0.7920; SNCA-/-: age factor F (1.20) = 0.03, *p* = 0.8615; SNCA-OVX: age factor F (1.23) = 0.002, *p* = 0.9634; Appendix A) and no differences between the genotypes (genotype factor F (2.71) = 0.25, *p* = 0.7827, Appendix A). Although the IL-4 levels appeared to be higher in the male than in the female mice, the differences did not reach statistical significance (wildtype: sex factor F (1.22) = 4.00, *p* = 0.0579; SNCA-/-: sex factor F (1.20) = 2.09, *p* = 0.1635; SNCA-OVX: sex factor F (1.23) = 3.42, *p* = 0.0773; Appendix A). Moreover, the quantification of the TNF-*α* revealed no differences with aging (wildtype: age factor F (1.23) = 2.08, *p* = 0.1631; SNCA-/-: age factor F (1.19) = 0.69, *p* = 0.4165; SNCA-OVX: age factor F (1.22) = 0.99, *p* = 0.3304; Appendix A), between the genotypes (genotype factor F (2.70) = 0.11, *p* = 0.8968, Appendix A), or between the sexes (wildtype: sex factor F (1.23) = 0.93, *p* = 0.3456; SNCA-/-: sex factor F (1.19) = 1.43, *p* = 0.2460; SNCA-OVX: sex factor F (1.22) = 2.32, *p* = 0.1420; Appendix A).

Finally, our results suggest a strong effect of aging on inflammatory process, including increases in astrogliosis, microgliosis, and some cytokine levels. However, the SNCA-OVX male mice seemed to present a higher increase in pro-inflammatory responses with aging compared to the other groups, specifically for astrogliosis and microgliosis.

## 4. Discussion

The present study used a progressive model of synucleinopathy to investigate whether female mice would be protected against αSyn-overexpression-induced PD, as observed in toxin-induced models [42,54]. For this purpose, we compared SNCA-OVX mice, a murine SNCA-gene-knockout model with the overexpression of h-αSyn, with both wildtype and SNCA-/- mice. We used SNCA-/- mice as controls, to compare the SNCA-knockout background present in both the SNCA-/- and the SNCA-OVX mice. The findings led us to conclude the following: both the accumulation of h-αSyn in the SNCA-OVX mice and the absence of αSyn expression in the SNCA-/- mice induced progressive motor and non-motor dysfunction. Only the SNCA-OVX mice displayed alterations in the nigrostriatal pathway, with a faster age-related loss of striatal DA content in the males. There was an overall age-related increase in pro-inflammatory responses for all the experimental groups. Astrogliosis and microgliosis were affected by biological sex in the SNCA-OVX model.

The first use of SNCA-OVX mice was by Janezic et al., as a model to study synucleinopathy. Constructed with a BAC carrying the wildtype human-SNCA locus, the protein was overexpressed at 1.9 times the endogenous-mouse-protein level in the transgenic mice. The authors used the rotarod performance and multiple static rods tests to show late motor deficits in 18-month-old, but not in 3-month-old mice [37]. Our study used the open field test to evaluate locomotor activity and anxiety-like behavior [55]. Firstly, we observed a significant progressive decrease in the total distance traveled by both the male and the female wildtype mice between the ages of 4 and 18 months. In wildtype male mice, the total distance traveled was significantly reduced between 4 and 12 months. On the other hand, both the SNCA-/- and the SNCA-OVX mice showed a significant progressive decrease in locomotor activity. Notably, the SNCA-OVX male mice showed a more significant decrease in distance traveled compared to the female mice between 4 and 12 months, suggesting a more abrupt loss of motor activity in the males around 12 months. By contrast, the reduction in distance traveled was more progressive in the female SNCA-OVX mice. Secondly, only the male wildtype mice presented a progressive increase in anxiety-like behavior with aging. The behavioral analyses of the SNCA-/- and SNCA-OVX mice revealed an age-dependent increase in anxiety that was significantly higher in both transgenic models than in the wildtype mice.

The overexpression of αSyn has been associated with both sporadic and familial forms of PD, emphasizing its importance in the neuropathology of the disease [3,56,57]. It was previously established that SNCA-OVX wildtype mice express almost twice as much h-αSyn as endogenous αSyn in mice without displaying protein-aggregation pathology [37,58]. Therefore, we investigated the possibility of age-related changes in h-αSyn levels in the SNCA-OVX mice and the likelihood of biological sex affecting the protein concentrations. Interestingly, we found that the h-αSyn levels increased in both males and females aged between 4 and 18 months. Moreover, our results showed a trend of higher levels of this protein in the females compared to the males, which could be explained by the capacity of the estrogen receptor alpha to modulate αSyn expression [59]. We studied the relevance of these results in our PD model by investigating the integrity of the nigrostriatal pathway.

Considerable evidence in PD models suggests that males are more susceptible to nigrostriatal pathology than females [60,61,62]. Studies with the MPTP-induced mouse model showed a substantial decrease in the DA transporter (DAT), the vesicular monoamine transporter 2 (VMAT2), and DA content, specifically in male mice [42,63]. However, the literature does not provide evidence on sexual differences in SNCA-overexpression models despite previous reports of motor symptoms with a loss of nigrostriatal DA neurons and abnormal striatal DA release in αSyn-based models [45,64,65]. In the first study, 18-month-old SNCA-OVX mice exhibited a 30% loss of TH+ neuron density in the SNpc and a deficit in striatal DA neurotransmission [37]. In accordance with these outcomes, our analysis of the DA cell bodies in the SNpc of the male and female SNCA-OVX mice showed decreases of 15% and 18%, respectively, from 4 to 18 months. Furthermore, the decrease in the striatal contents of DA, DOPAC, 3-MT, and HVA in both the female and the male SNCA-OVX mice between 4 and 18 months was in agreement with the observed loss of nigral DA neurons. This decrease in striatal DA contents can also be explained by the increased activity of DAT, facilitated by the αSyn [58]. Intriguingly, we observed that the DA contents and the TH+ cell counts were higher in the SNCA-OVX animals than in the wildtype and SNCA-/- mice at 4 months of age. We can speculate that the high expression levels of αSyn may have encouraged the genesis of DA neurons at an early developmental stage, prior to the initiation of neurodegenerative processes. Interestingly, in the SNCA-/- mice, the DA levels and TH+ cell counts were not different from those of the wildtype animals, suggesting that the overexpression of αSyn was involved in this early increase in DA cells and DA content. Moreover, similarly to the MPTP model of PD, biological sex differences may have influenced the loss of DA contents and some DA metabolites in the SNCA-OVX mice. Taken together, our results suggest that the loss of DA contents and motor activity is more progressive in female than in male SNCA-OVX mice. Supporting our present results, it has been shown that older female C57/BL6 mice lose their estrus cycle and neuroprotective female sex steroids with aging [66,67].

Observations of PD patients, toxin-based mouse models, and αSyn-based mouse models stress the importance of the immune system in the neuropathogenesis of PD [68,69,70,71]. Astrogliosis, microgliosis, and changes in cytokine expression are parts of the inflammatory processes [72]. Interestingly, we found an increase in astrogliosis with aging in both male and female SNCA-OVX mice, but no significant changes in the wildtype or SNCA-/- male mice. These results suggest a normal increase in astrogliosis with aging in male mice, along with a pro-inflammatory response specific to the overexpression of αSyn. Moreover, the significant age-related increase in astrogliosis in all the female mouse models in this study suggests that αSyn has no notable effect on astrocytes in females. Additionally, the effects of aging on astrogliosis specific to females were previously reported [73]. Analyses of microgliosis also revealed that the increase in the inflammatory response with aging in male mice was significant only in the SNCA-OVX types. Our results are in accordance with a previously reported increase in microgliosis in a Thy1-αSyn mouse model [21,74]. These increases in gliosis in synucleinopathy models can be explained by the possible interaction of αSyn with astrocytes and microglia [75]. Furthermore, several αSyn transgenic mouse models have also shown modulation in pro-inflammatory mediators through the activation of glial cells [76,77]. The quantification of plasma cytokines revealed increases in IL-1β, IL-5, IL-10, and IFN-γ with aging. Unlike those of IL-10 and IFN-γ, the levels of IL-1β and IL-5 were higher in the SNCA-OVX mice than in the wildtype mice. This suggests that in addition to the aging response, SNCA-OVX mice also display other pro-inflammatory markers. Taken together, these results suggest a strong genotype-independent immune response with aging and a specific increase in gliosis in SNCA-OVX male mice. Moreover, our results showed increased microgliosis in SNCA-OVX male mice with aging, whereas no increases were observed in the SNCA-OVX female mice. This is consistent with the results of our MPTP mouse model of PD, in which microgliosis was observed to be higher in the male MPTP mice than in the female mice [78].

The limitations of the present study include the use of only one behavioral test and the lack of characterization of other non-PD αSyn phenotypes. Moreover, gonadectomized mice were not investigated to unveil the role of sex hormones. This will be important to study in future experiments, considering that most women with PD are aged 50 years and older and in menopause, with an associated abrupt and significant loss of ovarian steroids, which can be modeled using gonadectomized female mice. Gonadectomized male mice could also be investigated to model andropause in men, but if decreases in sex steroids occur in men, they are more progressive during aging.

## 5. Conclusions

The present study revealed in mice overexpressing h-αSyn a more abrupt decrease in nigrostriatal DA content during aging and an increase in microgliosis in males but not in females. Human-αSyn overexpression was therefore toxic for the DA neurons. However, at 18 months of age, this sex difference in DA content was lost; this is likely to have been due to the decrease in ovarian functions, leading to reduced neuroprotection mediated by female sex steroids. Hence, female-hormone supplementation could be beneficial for PD patients when in the early stages of the disease.

## Figures and Tables

**Figure 1 biomolecules-13-00977-f001:**
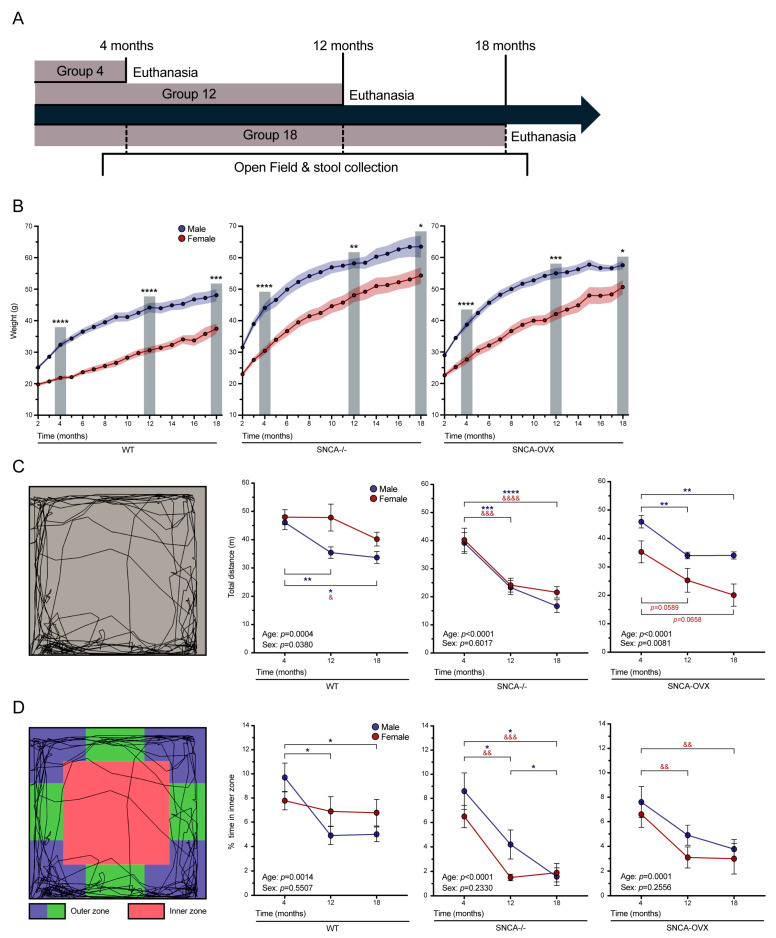
**Impact of *α*Syn overexpression on mouse weight and motor behavior**. (**A**) Timeline of experimentation, representing the groups euthanized at 4, 12, and 18 months of age. Open field experiments and stool collection were performed longitudinally. (**B**) Animals from the 18-month-group were weighed each month between 2 (weaning) and 18 months (euthanasia), showing that males were heavier than females and that all animals gained weight with age. (**C**) Analysis of the total distance traveled by mice in the open field test revealed a fast decrease in motor activities with age in SNCA-/- mice, with no sex differences. However, wildtype and SNCA-OVX mice showed sex differences, with male mice showing faster decreases than female mice. (**D**) Percentage of time in the open field inner zone made it possible to measure anxiety-like behavior. The fast decrease in the inner-zone percentage with SNCA-/- and SNCA-OVX mice showed an increase in anxiety with aging. Each group included 8–10 mice. Tukey’s post hoc tests: (male) * *p* < 0.05, ** *p* < 0.01, *** *p* < 0.001, **** *p* < 0.0001; (female) & *p* < 0.05, && *p* < 0.01, &&& *p* < 0.001, &&&& *p* < 0.0001. **Abbreviations:** WT, wildtype.

**Figure 2 biomolecules-13-00977-f002:**
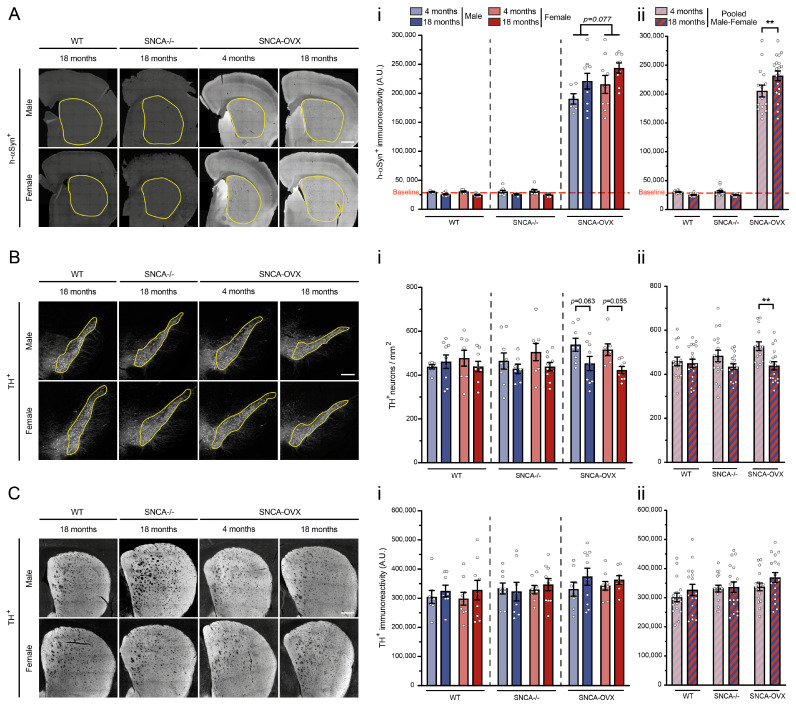
Accumulation of human *α*Syn led to the loss of nigral DA neurons. (**A**) Photomicrographs of h-*α*Syn immunoreactivity showing widespread expression in the brains of SNCA-OVX transgenic mice. Striatum contour is delineated with a yellow line. Scale bar = 500 µm. (**A**(**i**)) Immunoreactivity analysis of h-*α*Syn in the striatum showing a high concentration of the protein in the SNCA-OVX mice but none in the wildtype and SNCA-/- mice. (**A**(**ii**)) An increase in h-*α*Syn between 4 and 18 months can be observed, although it is only significant in sex-pooled SNCA-OVX mice. The baseline is the mean of 5 male and 5 female negative controls. (**B**) Photomicrographs of TH+ neurons in the SNpc, which are delineated with a yellow contour. Scale bar = 200 µm. (**B**(**i**)) Analysis of TH+ neuron density reveals a loss of DA cell bodies in the SNpc with age in both female and male SNCA-OVX transgenic mice. (**B**(**ii**)) A decrease in TH+ neuron density between 4 and 18 months can be observed, although it is only significant in sex-pooled SNCA-OVX mice. (**C**) Photomicrographs of TH+ neuronal projection in striatum. Scale bar = 400 µm. (**C**(**i**)) Immunoreactive quantification of TH+ neuronal projection in striatum revealed non-significant change in DA projections with age or sex. (**C**(**ii**)) No differences observed with age in sex-pooled mice. **Panel i**, results from male and female mice are not pooled. **Panel ii**, hatched bars, results from male and female mice are pooled. Each group included 7–10 mice. Bonferroni post hoc tests: ** *p* < 0.01. **Abbreviations:** h-*α*Syn, human *α*-synuclein; A.U., arbitrary unit; TH, tyrosine hydroxylase; WT, wildtype.

**Figure 3 biomolecules-13-00977-f003:**
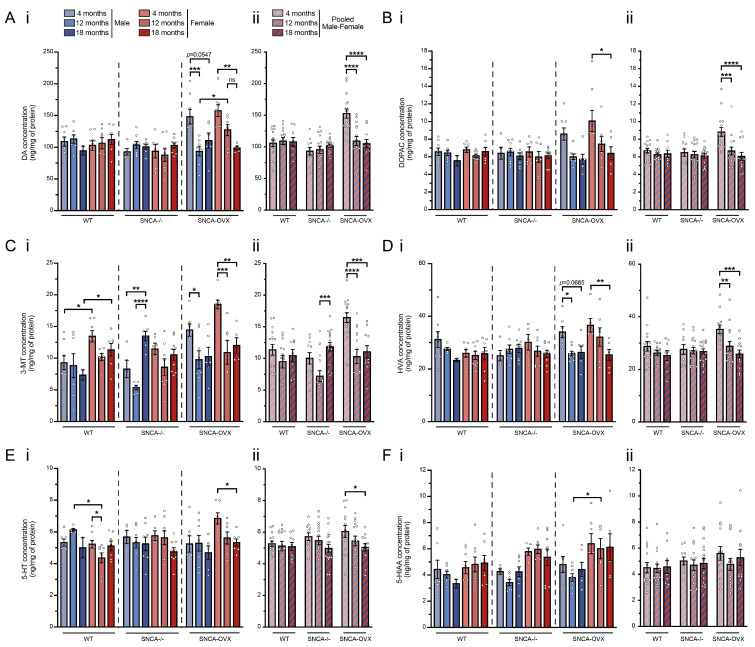
Overexpression of *α*Syn decreased striatal DA and its metabolite contents (**A**–**D**). Quantifications by HPLC of striatal DA (**A**) and its metabolite contents, (**B**) DOPAC, (**C**) 3-MT, and(**D**) HVA, revealing a significant decrease in DA and its metabolites with aging only in the SNCA-OVX mice (**E**,**F**). The striatal biogenic amine assay also showed an age-related loss of 5-HT in female SNCA-OVX mice (**E**) and higher levels of 5-HIAA in females compared to males in both SNCA-/- and SNCA-OVX mice (**F**). **Panel i**, results from male and female mice are not pooled. **Panel ii**, hatched bars, results from male and female mice are pooled. Each group included 6–9 mice. Bonferroni or Sidak post hoc tests: * *p* < 0.05, ** *p* < 0.01, *** *p* < 0.001, **** *p* < 0.0001. **Abbreviations**: 3-MT, 3-methoxytyramine; 5-HIAA, 5-hydroxyindoleacetic acid; 5-HT, 5-hydroxytryptamine; DA, dopamine; DOPAC, 3,4-Dihydroxyphenylacetic acid; HVA, homovanillic acid; WT, wildtype.

**Figure 4 biomolecules-13-00977-f004:**
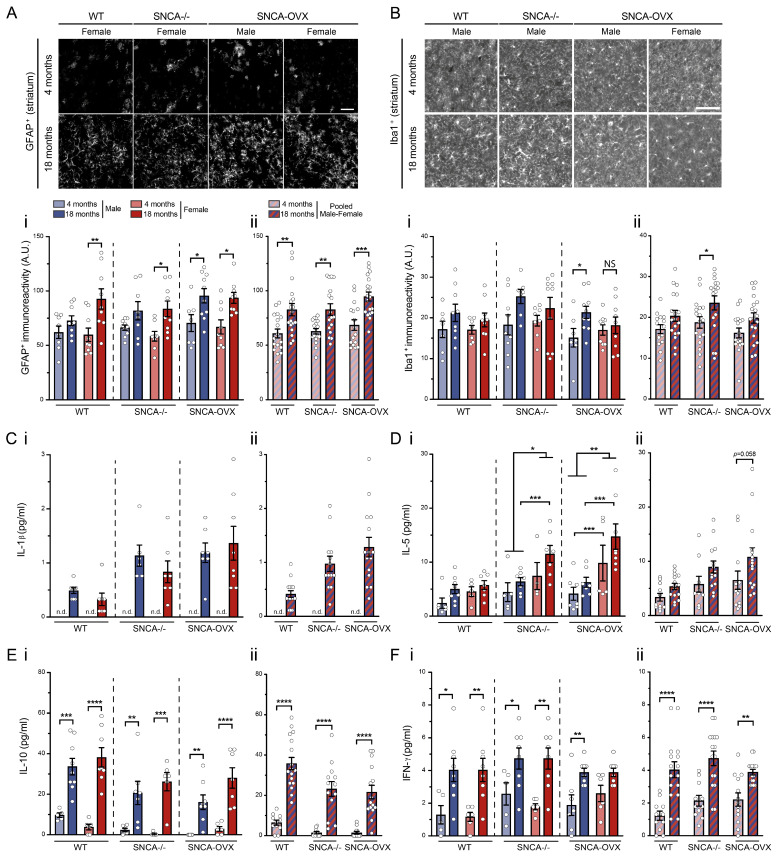
Inflammation is increased both by aging and by the h-*α*Syn increase. (**A**) Photomicrographs of astrocytes labeled with anti-GFAP antibody in the striatum. Scale bar = 50 µm. (**A**(**i**)) Quantification of GFAP+ astrocytes’ immunoreactivity showed an increase with aging in all female mice and an increase in SNCA-OVX male mice. (**A**(**ii**)) An increase in GFAP immunoreactivity was observed between 4 and 18 months in sex-pooled mice. (**B**) Photomicrographs of microglia cells labeled with anti-iba1 antibody in the striatum. Scale bar = 100 µm. (**B**(**i**)) Quantification of Iba1+ microglia by immunoreactivity revealed a significant increase in SNCA-OVX male mice, and a non-significant increase in the other male-mouse models. (**B**(**ii**)) When mouse models were sex-pooled, a significant increase in microgliosis with aging was observed in SNCA-/- animals only. (**C**–**F**). Plasma-cytokine analysis revealed an increase in IL-1*β* (**C**), IL-10 (**E**), and IFN-*γ* (**F**) with aging in both sexes. Quantification of IL-5 (**D**) showed no difference with aging, but a higher level in female compared to male mice. **Panel i**, results from male and female mice are not pooled. **Panel ii**, hatched bars, results from male and female mice are pooled. Each group included 7–10 mice. Bonferroni post hoc tests: * *p* < 0.05, ** *p* < 0.01, *** *p* < 0.001, **** *p* < 0.0001. NS, non-significant. **Abbreviations**: A.U., arbitrary unit; GFAP, glial fibrillary acidic protein; Iba1, ionized calcium-binding adapter molecule 1; IFN-*γ*, interferon gamma; IL, interleukin; WT, wildtype.

## Data Availability

The data presented in this study are available upon reasonable request from the corresponding authors.

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
