# Peer review of "Sex and Age Differences in a Progressive Synucleinopathy Mouse Model"

_biomolecules, 2023, doi:10.3390/biom13060977_

Round 1
Reviewer 1 Report
I think some figures need to be better presented, on the photomicrographs it is difficult to see what has been described in the text. It is preferable to enter magnifications
In line 124 the size of the measure is wrong
Author Response
Comment 1 : I think some figures need to be better presented, on the photomicrographs it is difficult to see what has been described in the text. It is preferable to enter magnifications
Answer 1 : Figure legends were improved. Scale bars have been better described.
Comment 2 : In line 124 the size of the measure is wrong
Answer 2 : done
Reviewer 2 Report
The study by Lamontagne-Proulx et al. used synuclein (aSyn) mice, including aSyn overexpressing mice (SNCA-OVX), SNCA knockout mice, and wild-type mice, to investigate the effects of aSyn. The authors examined differences in body weight, locomotion (using an open field test), aSyn expression in various brain regions, dopamine (DA) and related metabolite levels, and inflammation. Although the authors claimed to investigate sex differences in their synucleinopathy model, the results do not show any differences between males and females. Instead, the results suggest an aging effect. Therefore, it is unclear what the point of the study was, and it is not suitable for publication in the Biomolecules journal.
Many concerns with the study include:
1. The authors aimed to investigate sex differences, but the data were compared in terms of age instead of sex.
2. To investigate synucleinopathy, the authors should have used a more detailed motor function behavior study or a study specifically for Parkinson's disease instead of just an open-field assay.
3. The authors did not observe hormone differences among males, females, and ages.
4. In Figure 2, the authors should have used a general anti-aSyn antibody to detect different levels of aSyn.
5. The description of Figure 2 is too brief, and it is unclear what the authors wanted to convey. Additionally, there were no differences in the effect of sex.
6. The quality of Figures 4A and 4B is poor, and it is unclear what the aspect of sex difference is.
7. There is a Typo in P6 line 224
Author Response
Many concerns with the study include:
- The authors aimed to investigate sex differences, but the data were compared in terms of age instead of sex.
Answer: The present study investigated mice of three different genotypes and for each genotype we investigated disease progression in males and females at three ages. Some comparisons of sex differences were included in the original submission. This is now completed with additional male/female statistical comparisons as well as better described and emphasized in the graphs.
- To investigate synucleinopathy, the authors should have used a more detailed motor function behavior study or a study specifically for Parkinson's disease instead of just an open-field assay.
Answer: the SNCA-OVX mice were previously very well characterized in detail be the authors that generated these transgenic animals. This was reported in a very high impact factor paper in PNAS in 2013. Hence, we chose to investigate only one motor and one anxiety test.
- 3. The authors did not observe hormone differences among males, females, and ages.
Answer: Since we used female mice at random stages of the estrous cycle we chose not to investigate the individual sex steroids in detail but measure their combined effect on behavior, DA systems and inflammation. A next step in the present studies would include the investigation of gonadectomized mice to model menopause and andropause. This has been added in the discussion as a limitation of the present study.
- In Figure 2, the authors should have used a general anti-aSyn antibody to detect different levels of aSyn.
Answer: Since SNCA-OVX mice overexpress human α-synuclein and do not express mouse α-synuclein, we measured only the human form of α-synuclein in the present study.
- The description of Figure 2 is too brief, and it is unclear what the authors wanted to convey. Additionally, there were no differences in the effect of sex.
Answer: Legends to Figure 2 has been rewritten to describe better the experimental results in the different panels. Indeed, there was no sex-differences in h-Syn expression, TH+ neuron density in the SNpc and striatal TH immunoreactivity.
- The quality of Figures 4A and 4B is poor, and it is unclear what the aspect of sex difference is.
Answer: Figure 4 was improved. Microglia were shown at higher magnification and additionnal statistical comparisons were included.
- There is a Typo in P6 line 224
Answer : done
Reviewer 3 Report
Using a synucleinopathy mouse model, Lamontagne-Proulx et al. addressed the impact of sex on αSyn-related PD phenotypes, including motor functions, loss of nigral DA neurons and striatal biogenic amines, and onset of pro-inflammatory events in age-dependent manner. Specifically, the authors investigated these molecular and behavioral outcomes in SNCA-OVX transgenic mice, which overexpress human wildtype α-synuclein and exhibit age-dependent loss of nigrostriatal dopamine neurons and motor deficits characteristic of PD. Surprisingly, in the paper in which this animal model has been characterized for the first time, Janezic and collaborators (2003, doi: 10.1073/pnas.1309143110) focused their investigations mainly in male SNCA-OVX mice (they only compared gastro-intestinal functions between males and females). Here, Lamontagne-Proulx and collaborators, examine the abovementioned PD-like features in male and female mice. A further strength of the manuscript is the comparison between SNCA-OVX mice with SNCA-/- mice and wild type. Overall, the authors provide evidence that both the accumulation and lack of αSyn expression elicited progressive motor and non-motor dysfunctions; conversely, only SNCA-OVX mice exhibited alterations in the nigrostriatal pathway, and male counterparts showed a more robust loss of motor activity. While the authors found an overall age-related increase in pro-inflammatory responses for all experimental groups, the astrogliosis and microgliosis of SNCA-OVX were sex biased.
Overall, the manuscript is well-written, the methods are sound and the results are clearly presented. The experimental design and statistical analyses appear appropriate, with the inclusion in each experimental design of the variable sex, time, and wild type as the control group.
I only have a few minor suggestions the authors may wish to address. Please see below.
The title should be more specific, in this form is too general.
The abstract needs some changes. First, better specify what SNCA-OVX and SNCA-/- stand for. Some readers might be not familiar with the animal model. Second, the conclusion does not sufficiently address the results.
Please rephrase the hypothesis formulated in the introduction. The authors “hypothesize that the accumulation of αSyn causes progressive motor impairments and cognitive deficits”. It is well established that the accumulation of αSyn causes progressive motor impairments and cognitive deficits. Also, it has been already documented that these transgenic mice show PD-like phenotype in the nigrostriatal pathway, such as loss of both DA neurons and DA contents, as well as an increase in pro-inflammatory response. I assume that the hypothesis might be that female and male SNCA-OVX mice might display sex differences in the manifestation of these phenotypes. Therefore, this animal model could help understand the role of sex and sex hormones in the sequence of events leading to PD. Likewise, please add the aim of the study to the introduction.
Please add some limitations of the study to the discussion section (for instance, only one behavioral test, the lack of characterization of other non-PD alpha-synuclein phenotypes; no investigations on gonadectomized animals to unveil the role of sex hormones).
While not mandatory, I suggest adding a conclusion section in order to summarize the main findings and mention the novelty of the study.
Please also:
- Change “gold standard” with “hallmark” (line 38)
- Change “efficacity” with "efficacy" or "ability" (line 73)
- add the third person to “support” or change with “there is little evidence” supporting… (line 76)
- Delete from the sentence the redundant term “this model” (line 83)
- Delete “they” or rephrase (for instance: In addition, these mice exhibit…) (line 84)
- Delete “conversely” from the sentence (line 210)
There is a strange symbols for “alpha” in alpha-synuclein throughout the text. Please check.
The English language is fine, with minor issues.
Author Response
General comment: Using a synucleinopathy mouse model, Lamontagne-Proulx et al. addressed the impact of sex on αSyn-related PD phenotypes, including motor functions, loss of nigral DA neurons and striatal biogenic amines, and onset of pro-inflammatory events in age-dependent manner. Specifically, the authors investigated these molecular and behavioral outcomes in SNCA-OVX transgenic mice, which overexpress human wildtype α-synuclein and exhibit age-dependent loss of nigrostriatal dopamine neurons and motor deficits characteristic of PD. Surprisingly, in the paper in which this animal model has been characterized for the first time, Janezic and collaborators (2003, doi: 10.1073/pnas.1309143110) focused their investigations mainly in male SNCA-OVX mice (they only compared gastro-intestinal functions between males and females). Here, Lamontagne-Proulx and collaborators, examine the abovementioned PD-like features in male and female mice. A further strength of the manuscript is the comparison between SNCA-OVX mice with SNCA-/- mice and wild type. Overall, the authors provide evidence that both the accumulation and lack of αSyn expression elicited progressive motor and non-motor dysfunctions; conversely, only SNCA-OVX mice exhibited alterations in the nigrostriatal pathway, and male counterparts showed a more robust loss of motor activity. While the authors found an overall age-related increase in pro-inflammatory responses for all experimental groups, the astrogliosis and microgliosis of SNCA-OVX were sex biased.
Overall, the manuscript is well-written, the methods are sound and the results are clearly presented. The experimental design and statistical analyses appear appropriate, with the inclusion in each experimental design of the variable sex, time, and wild type as the control group.
Answer to general comment : We appreciate these positive comments on our manuscript.
I only have a few minor suggestions the authors may wish to address. Please see below.
Comment: The title should be more specific, in this form is too general.
Answer : we modified the title according to this comment.
The abstract needs some changes. First, better specify what SNCA-OVX and SNCA-/- stand for. Some readers might be not familiar with the animal model. Second, the conclusion does not sufficiently address the results.
Answer: done as suggested
Please rephrase the hypothesis formulated in the introduction. The authors “hypothesize that the accumulation of αSyn causes progressive motor impairments and cognitive deficits”. It is well established that the accumulation of αSyn causes progressive motor impairments and cognitive deficits. Also, it has been already documented that these transgenic mice show PD-like phenotype in the nigrostriatal pathway, such as loss of both DA neurons and DA contents, as well as an increase in pro-inflammatory response. I assume that the hypothesis might be that female and male SNCA-OVX mice might display sex differences in the manifestation of these phenotypes. Therefore, this animal model could help understand the role of sex and sex hormones in the sequence of events leading to PD. Likewise, please add the aim of the study to the introduction.
Answer: done as suggested.
Please add some limitations of the study to the discussion section (for instance, only one behavioral test, the lack of characterization of other non-PD alpha-synuclein phenotypes; no investigations on gonadectomized animals to unveil the role of sex hormones).
Answer: done as suggested
While not mandatory, I suggest adding a conclusion section in order to summarize the main findings and mention the novelty of the study.
Answer: done
Please also:
- Change “gold standard” with “hallmark” (line 38)
- Change “efficacity” with "efficacy" or "ability" (line 73)
- add the third person to “support” or change with “there is little evidence” supporting… (line 76)
- Delete from the sentence the redundant term “this model” (line 83)
-
- Delete “conversely” from the sentence (line 210)
Answer: done
There is a strange symbols for “alpha” in alpha-synuclein throughout the text. Please check.
Answer: done
Reviewer 4 Report
In this manuscript, the authors provide a comprehensive overview of the alpha-synuclein protein linked to Parkinson's disease and underpinning the potential sex differences using a mouse model.
The manuscript is well-written, and clearly explains the researchers' findings. The originality of the subject is that contrasting males and females enable us to learn more about Parkinson's disease. Although using females at different estrous cycles rather than an ovariectomized model would be interesting, the manuscript could be accepted in its current form after one more revision to fix any text errors.
Author Response
In this manuscript, the authors provide a comprehensive overview of the alpha-synuclein protein linked to Parkinson's disease and underpinning the potential sex differences using a mouse model.
The manuscript is well-written, and clearly explains the researchers' findings. The originality of the subject is that contrasting males and females enable us to learn more about Parkinson's disease. Although using females at different estrous cycles rather than an ovariectomized model would be interesting, the manuscript could be accepted in its current form after one more revision to fix any text errors.
Answer : we included in the limitations of the present study that ovariectomized female mice should be used in future experiments to unveil the role of sex steroids.